# The Effect of Continuous Intake of *Lactobacillus gasseri* OLL2716 on Mild to Moderate Delayed Gastric Emptying: A Randomized Controlled Study

**DOI:** 10.3390/nu13061852

**Published:** 2021-05-28

**Authors:** Toshihiro Ohtsu, Ken Haruma, Yumiko Ide, Atsushi Takagi

**Affiliations:** 1Food Microbiology Research Laboratories, Applied Microbiology Research Department, Meiji Co. Ltd., 1-29-1 Nanakuni, Hachiouji, Tokyo 192-0919, Japan; 2General Medical Center, Department of Internal Medicine 2, Kawasaki Medical School, 577 Matsushima, Kurashiki, Okayama 701-0192, Japan; kharuma@med.kawasaki-m.ac.jp; 3Doctor’s Office, Tokyo Center Clinic, 1-1-8 Yaesu, Chuo-ku, Tokyo 103-0028, Japan; ide@tc-clinic.jp; 4Department of Internal Medicine, Tokai University School of Medicine, Shimokasuya, Isehara, Kanagawa 259-1193, Japan; ta60264@tsc.u-tokai.ac.jp

**Keywords:** probiotics, gastric motility, ^13^C gastric emptying breath test, gastric emptying rate, gastroparesis, autonomic nerves system

## Abstract

Probiotics have been suggested to be effective for functional dyspepsia, but their effect on gastric motility is not clear. We evaluated the effect of *Lactobacillus gasseri* OLL2716 (LG21 strain) on mild to moderate delayed gastric emptying by a double-blind, parallel-group, placebo-controlled, randomized trial. Participants (*n* = 28) were randomly assigned to ingest LG21 strain-containing yogurt (LG21 strain group) or LG21 strain-free yogurt (placebo group) for 12 weeks. The ^13^C gastric emptying breath test was performed to measure the gastric emptying rate over time following ingestion of a liquid meal, and the time to reach the peak (Tmax) was used as an indicator of gastric emptying. We also measured the salivary amylase concentration, an indicator of autonomic dysfunction under stress. The per-protocol population (*n* = 27, male *n* = 4, female *n* = 23) was evaluated for efficacy. When a ≥30% reduction in the difference between participant’s Tmax and the Japanese mean Tmax was defined as an improvement, the odds ratio of improvement in delayed gastric emptying compared to placebo after 12 weeks was 4.1 (95% confidence interval, 0.8 to 20.2). Moreover, salivary amylase concentrations were significantly lower than in the placebo group, indicating an improvement in autonomic function. The present data were not enough to support the beneficial effects of the LG21 strain on delayed gastric emptying. However, if we define the odds ratio in further study investigated with a larger number of participants, LG21 strain might be expected to have some impact on delayed gastric emptying.

## 1. Introduction

When a meal is ingested, the fundus of the stomach actively relaxes and accepts the ingested food, followed by contraction of the antrum to mash and expel the chyme from the pylorus into the duodenum (gastric emptying). Gastroparesis is a disorder characterized by severe delayed gastric emptying (i.e., a long time is required for emptying the gastric contents), whose main symptoms include postprandial fullness, early satiety, nausea, belching, and abdominal bloating [1,2]. Although gastroparesis is estimated to affect only 1.3% to 1.4% of the general population [2], many others experience mild to moderate delayed gastric emptying, which is one of the most common causes of stomach upset. Although the pathological mechanism of delayed gastric emptying is not fully understood, the autonomic nervous system plays a significant role in gastric motility. In particular, the vagus nerve has excitatory and inhibitory effects due to its postganglionic action [3,4], and the vagus nerve action is transmitted to the smooth muscles by the interstitial cells of Cajal (ICC) in the smooth muscle layer of the stomach. In recent years, the reduction in or damage to ICCs and nerves has been revealed in gastroparesis, which is now discussed as a part of the broader spectrum of gastric neuromuscular dysfunction [1]. There is also growing attention on the immune system abnormalities that have been revealed to be deeply involved in these neuromuscular dysfunctions. Specifically, a decrease in anti-inflammatory macrophages, which protect nerves from oxidative stress and inflammation, was shown in association with gastroparesis [5]. About one-third of the patients with gastroparesis are diabetic, but most cases of gastroparesis are idiopathic [6]. In addition to abnormalities in glucose metabolism due to dietary disturbances, disorders of the autonomic nervous system due to stress and lifestyle disturbances may cause delayed gastric emptying [7,8].

Although the effect of probiotics on gastric motility is unclear, we reported in a randomized controlled trial of *L. gasseri* OLL2716 (LG21 strain) in *H. pylori*-uninfected functional dyspepsia (FD), that continuous LG21 strain intake is effective in postprandial distress syndrome (PDS), a subtype of FD [9]. Since prokinetic agents have been reported to be effective in PDS [10], the LG21 strain may be effective in delayed gastric emptying. It has also been reported that the LG21 strain may improve autonomic nervous system disorders [11], and it may therefore similarly improve delayed gastric emptying. In addition to this evidence, the LG21 strain can be used as a food, which means that it can be ingested daily at the individual’s discretion without relying on medical facilities, making it suitable for health maintenance and primary care. Therefore, we performed a double-blind, parallel-group, placebo-controlled, randomized controlled trial (RCT) of the LG21 strain in individuals with mild to moderate delayed gastric emptying as assessed by breath test to investigate the effect of continuous intake of the LG21 strain on gastric motility.

## 2. Materials and Methods

### 2.1. Study Design

In this double-blind, parallel-group, placebo-controlled RCT, participants were assigned to either the LG21 strain group or placebo group (Figure 1). The study was performed from November 2018 to May 2019 at the Tokyo Center Clinic (Tokyo, Japan). This study was approved by the ethics committee of Meiji Co., Ltd. (Tokyo, Japan) and the Japan Conference of Clinical Research (Tokyo, Japan) following a review of its scientific and ethical validity. The purpose and content of this study were explained to the participants before they participated in this study in accordance with the principles of the Declaration of Helsinki and the ethical guidelines for epidemiological research. All participants provided written informed consent prior to participation in this study. The study protocol was registered with the University Hospital Medical Information Network (UMIN) Clinical Trial Registry (UMIN000035057) on 31 August 2019.

### 2.2. Participants

Healthy Japanese individuals aged between 20 and 64 years, who experienced stomach upset during the past month for which they did not receive any treatment, were recruited for the study through advertisements on a website managed by a participant recruitment company (3H Clinical Trial Inc.; Tokyo, Japan). At screening, recruited individuals were confirmed to have occasionally experienced mild (symptoms easily tolerated) to moderate-to-severe (symptoms occasionally limited daily activities) stomach upset during the past month as well as to have delayed gastric emptying as assessed by a ^13^C gastric emptying breath test with a time to reach the peak gastric emptying rate (Tmax) of more than 55 min after ingestion of a liquid meal. Eligibility was determined by whether individuals met the inclusion criteria and did not meet the exclusion criteria. The exclusion criteria were as follows: severe delayed gastric emptying (Tmax ≥ 75 min) and severe heartburn or reflux symptoms at the time of screening; visits to a medical institution for disorders of the gastrointestinal tract or diabetes during the 6-month period prior to the study; suspected diabetes, disorders of the gastrointestinal tract, or severe renal impairment based on the blood tests (hematology and biochemistry), physical examination, and history taken at the time of screening; bothersome gastric discomfort persisting for more than 6 months and occurring regularly for the 3-month period prior to the study (suspected FD); *H. pylori* positive by blood test at the time of screening or history of *H. pylori* eradication therapy; and any other individual who was judged by the investigator to be inappropriate for the study.

### 2.3. Criteria for Tmax at Screening

Individuals with a Tmax greater than 55 min and less than 75 min were included in the study. These criteria correspond to the mean Tmax for healthy Japanese individuals +1 standard deviation (SD) or more but less than +3 SD. Specific values were based on a survey report from the Japan Society of Smooth Muscle Research (mean Tmax and SD were 45 min and 10 min, respectively) [12]. Although there is no international consensus on the criteria for gastroparesis, the need for a strict definition has been acknowledged. For example, +3 SD above the mean for healthy adults has been proposed [13,14]. In this study, participants were selected to exclude those with +3 SD or more as gastroparesis, while those with +1 SD or more and less than +3 SD were included as participants with mild to moderate delayed gastric emptying.

### 2.4. Study Protocol

The pooled participants were randomly and collectively assigned to one of two groups by one person responsible for the randomization by using a computer-generated random sequence. Stratified randomization was performed to balance gender and figure based on body mass index (BMI) across the two groups. Participants in the placebo group were asked to ingest one unit (85 g) of yogurt composed of a mixture of raw milk, dairy products, sugar, a sweetener (stevia), and raw water, and fermented with *Lactobacillus delbrueckii* subsp. *bulgaricus* and *Streptococcus thermophilus*. Participants in the LG21 strain group were asked to ingest yogurt containing the LG21 strain, added to the same yogurt described above (with ≥10^9^ colony-forming units (CFU) of the LG21 strain per yogurt unit). Both the placebo and the LG21 strain-containing yogurt contained the following nutrients per 85 g unit: energy, 68 kcal; protein, 2.9 g; lipids, 2.6 g; carbohydrates, 8.3 g; sodium, 37 mg; and calcium, 102 mg. Based on flavor and external appearance, it was confirmed that the test food products were indistinguishable. Participants were asked to ingest one unit (85 g) per day of the test food product for a period of 12 weeks. Participants were free to ingest it at any time of the day. In addition to the person in charge of allocation, who assigned the participants to two groups and determined the group number of the participants, the person who decided the identification number of the test food products, and the person who assigned the identification number of the test food products to the group number were separated respectively. As a result, the participants and all study staff were blinded to the test food by using the method in which the three pieces of information necessary for the key opening were not collected.

### 2.5. Assessments

#### 2.5.1. ^13^C Gastric Emptying Breath Test

A ^13^C gastric emptying breath test was performed before test food intake and after 6 and 12 weeks of test food intake using a liquid meal. One hundred milligrams of ^13^C-acetate (−^13^COOH; sodium) labeled with the stable isotope ^13^C (atomic weight 13) was mixed into a 200 mL/200 kcal liquid meal (Ensure Liquid^®^, Abbott, Chicago, IL, USA). Exhaled air was collected multiple times from 5 min before to 90 min after ingestion of the liquid meal, and the ^13^CO_2_:^12^CO_2_ ratio was measured using an infrared spectrophotometer (POCone^®^, Otsuka Electronics Co., Ltd., Osaka, Japan). The rate of gastric emptying into the duodenum (% dose/h) at the time of exhaled air collection was calculated by measuring the change in the ^13^CO_2_:^12^CO_2_ ratio (Δ‰) of exhaled air collected after ingestion of the liquid meal compared to the baseline ^13^CO_2_:^12^CO_2_ ratio before ingestion of the liquid meal, and then approximating the amount of CO_2_ produced in the body per hour by body surface area × 300 [15]. The time to reach the peak gastric emptying rate (Tmax) was used as an indicator of gastric emptying. The liquid meal was ingested at 10:30 a.m. on each test day. The participants were required to fast from 9:00 p.m. on the day before the test until completion of the test but were allowed to drink water until 8:00 a.m. on the day of the test. The ^13^C-acetate emptied from the stomach with the liquid meal is rapidly excreted in exhaled air as ^13^CO_2_ through absorption in the duodenum and metabolism in the liver. Although this test method has many steps, it has been reported to correlate well with direct scintigraphic results when using a liquid meal [16,17].

#### 2.5.2. Salivary Amylase Test

The concentration of amylase (U/mL) in saliva was measured before test food intake and after 6 and 12 weeks of test food intake using the α-amylase assay kit (CicaLiquid-N AMY; Kanto Chemical Co., Inc., Tokyo, Japan). Prior to the breath test on test visit, the participants were asked to place cotton wool in their mouth for the collection of saliva.

#### 2.5.3. Participant’s Global Assessment of Gastric Condition

After 6 and 12 weeks of test food intake, a questionnaire determining the participants’ impression of the overall effect on the gastric condition was completed. The following question was asked to the participants: “How have your overall gastric symptoms been during the past week, compared with your symptoms during the baseline period preceding the intake of the test food?” Responses were given based on the 7-point Likert scale [18]: (1) extremely improved, (2) improved, (3) slightly improved, (4) no change, (5) slightly aggravated, (6) aggravated, and (7) extremely aggravated.

#### 2.5.4. Endpoints

Primary endpoint: An intergroup comparison of the ΔTmax-area under the curve (AUC) during the intake period was performed. The ΔTmax-AUC of each participant during the intake period was obtained by plotting the origin, ΔTmax after 6 weeks of intake, and ΔTmax after 12 weeks of intake on a graph with intake time (weeks) on the *x*-axis and ΔTmax (minutes) on the *y*-axis, and connecting each plot with a straight line to calculate the AUC relative to baseline.Secondary endpoints: (1) Changes in salivary amylase concentration (Δsalivary amylase concentration) after 6 and 12 weeks of intake were compared to before intake. (2) A global assessment of gastric condition after intake was conducted.

Three other post hoc endpoints, all of which are the basic endpoints needed to assess the effectiveness of the intervention, were assessed as follows (planned after unblinding): Tmax after 6 and 12 weeks of intake compared to before intake in each group, the odds ratio for improvement in delayed gastric emptying in the LG21 strain group compared to the placebo group after 12 weeks of intake, gastric emptying rate following ingestion of a liquid meal at the time of gastric emptying breath test performed after 12 weeks of intake. In addition, exploratory analysis of confounding factors was performed on the relationship between changes in salivary amylase concentration after 12 weeks of intake and baseline salivary amylase concentration, and the effect of gender on the results of the intervention in delayed gastric emptying.

#### 2.5.5. Safety Assessment

For each group of participants who had ingested the test food product at least once, an assessment was made of any side effects and/or adverse events that had occurred during the study period.

### 2.6. Statistical Analysis

We previously reported that the LG21 strain is effective for FD (especially PDS) [9]. The number needed to treat (NNT) for the LG21 strain calculated from individuals who exhibited improvement based on the impression of the overall effect was 6.5 (the difference in the percentage of responders between the placebo and the LG21 strain groups was 15.5%). Although the result of 12 weeks of continuous intake, the NNT of the LG21 strain for FD was large and comparable to that of prokinetic agents [18]. Therefore, assuming an effect size based on the mean difference (Cohen’s *d*) of 0.8, a sample size of 20 per group was estimated to be sufficient to detect a statistically significant difference between the groups, with a power of 70% and a significance level of 5%.

Intergroup comparisons at baseline were performed using the Fisher’s exact test for categorical variables and the unpaired Student *t-*test or Wilcoxon rank-sum tests for continuous variables. Statistical analysis of the primary endpoint was performed using the Wilcoxon rank-sum test. Intergroup comparison of the Δsalivary amylase concentration was also performed using the Wilcoxon rank-sum test. Statistical analysis of the impression score of the overall effect on gastric condition after 12 weeks of intake was performed using ordinal logistic regression analysis. In addition, for the post hoc endpoints, statistical analysis of Tmax after intake compared to before intake was performed using the Friedman test, the odds ratio for improvement in delayed gastric emptying in the LG21 strain group compared to the placebo group was analyzed using logistic regression analysis, and the gastric emptying rate following ingestion of a liquid meal at the gastric emptying breath test performed after 12 weeks of intake was analyzed using two-way repeated-measures analysis of variance. In addition, as an exploratory investigation of confounding factors, analysis of variance was performed to investigate the relationship between changes in salivary amylase concentration after 12 weeks of intake and baseline salivary amylase concentration. Binomial logistic regression analysis was performed to investigate the effect of gender on the results of the intervention in delayed gastric emptying. The level of significance was set at 5% on both sides. In accordance with the pre-specified protocol, an analysis of efficacy was conducted on the per-protocol (PP) population.

## 3. Results

### 3.1. Participant Selection and Baseline Characteristics

A total of 298 Japanese individuals (149 male, 149 female), aged between 20 and 64 years, who had stomach upset and did not receive any treatment, were recruited and selected by means of two screening tests (Figure 2). A total of 28 participants (4 male, 24 female) were judged to meet the inclusion criteria. We were, therefore, unable to secure a target sample size of 40. These 28 individuals were randomized to each group (14 individuals per group), and all participants completed the study. However, one of those who completed the study was found to have violated the inclusion criteria (medication for sleep disorder during screening). Therefore, the PP population consisted of 27 individuals (14 individuals in the placebo group and 13 individuals in the LG21 strain group). As shown in Table 1, no significant between-group differences were found in the baseline characteristics of participants in the ITT population (4 men and 24 women, mean age 41.1 ± 11.0 years).

### 3.2. Primary Endpoint

The mean ± standard deviation of ΔTmax-AUC during the intake period was −7.5 ± 106.4 in the placebo group and −25.4 ± 55.9 in the LG21 strain group. Thus, there was greater variance in the placebo group than in the LG21 strain group, and homoscedasticity was significantly rejected (*p* < 0.05). The intergroup comparison of ΔTmax-AUC during the intake period using nonparametric tests showed no statistically significant difference (Figure 3a).

### 3.3. Secondary Endpoints

#### 3.3.1. Participant’s Global Assessment of Gastric Condition

There were no statistically significant between-group differences with respect to participants’ impression scores of the overall effect on gastric condition (Figure 3b).

#### 3.3.2. Δ Salivary Amylase Concentration after 12 Weeks of Intake

There was a statistically significant decrease in Δsalivary amylase concentration in the LG21 strain group compared to that in the placebo group after 12 weeks of intake (*p* < 0.05, Figure 3d).

#### 3.3.3. Tmax after 6 and 12 Weeks of Intake Compared to Before Intake

The difference was not statistically significant, although Tmax after 6 and 12 weeks of intake decreased in the LG21 strain group compared to before intake (*p* < 0.10, Figure 4).

#### 3.3.4. Odds Ratio for Improvement in Delayed Gastric Emptying after 12 Weeks of Intake

When a ≥30% reduction in the difference between participant’s Tmax and the Japanese mean Tmax (which is 45 min) was defined as an improvement in delayed gastric emptying, the numbers of participants with improvement and without improvement were 5 and 9 in the placebo group (improvement odds, 0.6), respectively, and 9 and 4 in the LG21 strain group (improvement odds, 2.3), respectively, after 12 weeks of test food intake. The odds ratio for improvement in delayed gastric emptying in the LG21 strain group compared to that in the placebo group after 12 weeks of intake was 4.1 (95% CI, 0.8 to 20.2).

#### 3.3.5. Gastric Emptying Rate following Ingestion of a Liquid Meal after 12 Weeks of Intake

The difference was not statistically significant, although the gastric emptying rate following ingestion of a liquid meal after 12 weeks of intake was suppressed in the LG21 strain group compared to that in the placebo group during the entire duration of 90 min of measurement (group effect, *p* < 0.10; Figure 3c).

#### 3.3.6. Relationship between Changes in Salivary Amylase Concentration after 12 Weeks of Intake and Baseline Salivary Amylase Concentration

Simple linear regression analysis was performed in which the dependent variable was post-intervention Δsalivary amylase concentration and the independent variable was baseline salivary amylase concentration. A model yielding significant results (R2 = 0.745, *p* < 0.001) was achieved in the LG21 strain group, showing a negative correlation with a regression coefficient of –0.42 (95% CI, –0.59 to –0.26) (Figure 5). On the other hand, there were no significant results in the placebo group (R2 = 0.012, *p* = 0.713). Analysis of variance was performed using the dependent variable of Δsalivary amylase concentration and the independent variables of the group, baseline salivary amylase concentration (Log), and interaction. There were statistically significant differences in group and interaction between the group and baseline salivary amylase concentration (*p* = 0.039 and *p* = 0.029, respectively).

#### 3.3.7. Effect of Gender on the Results of the Intervention in Delayed Gastric Emptying

Binomial logistic regression analysis was performed using the dependent variable of the presence or absence of improvement in delayed gastric emptying, and the independent variables of the group, gender, and interaction. There was no statistically significant difference in the interaction between the group and gender (*p* = 0.446).

### 3.4. Safety Assessment

A safety assessment was performed for 28 participants who had ingested the test food product at least once. Thirteen adverse events were reported in the placebo group (six cases of cold, three cases of gastrointestinal symptoms, three cases of sleep disorders, and one case of arthralgia) and 19 in the LG21 strain group (six cases of cold, six cases of gastrointestinal symptoms, four cases of sleep disorder, two cases of headache, and one case of arthralgia), all of which were non-serious, transient, and resolved. None of the adverse events caused problems with respect to the continued intake of the test food. There was no causal relationship between the test food and the adverse events, and none of the adverse events were side effects.

## 4. Discussion

From among healthy Japanese aged 20–64 years who had experienced gastric upset in the past month but had not been treated, we selected those with mild to moderate delayed gastric emptying as confirmed by breath testing. The RCT study was then carried out on those selected individuals, and the effects of continuous intake of the LG21 strain on gastric motility were investigated. In the study, participants were assigned to ingest either LG21 strain-containing yogurt or LG21 strain-free yogurt (placebo group), and changes in the groups due to continuous intake were compared. From this, the particular effectiveness of continuous intake of the LG21 strain was assessed. The results of the PP population analysis showed that there was no statistically significant difference between the groups with respect to ΔTmax-AUC, an indicator of the changes in gastric emptying during the intake period (the primary endpoint) (Figure 3a). However, the variance of the placebo group was greater than that of the LG21 strain group, and homoscedasticity was significantly rejected, suggesting that the participants who ingested the placebo yogurt were divided into those with and without improvement in delayed gastric emptying. When a ≥30% reduction in the difference between participant’s Tmax and the Japanese mean Tmax (which is 45 min) was defined as an improvement in delayed gastric emptying, the odds ratio for improvement in delayed gastric emptying in the LG21 strain group compared to the placebo group after 12 weeks of intake was 4.1 (95% CI, 0.8 to 20.2). On the other hand, it is known that the autonomic nervous system plays a major role in gastric motility. In the PP population, salivary amylase concentration, an indicator of autonomic dysfunction under stress [19,20], was significantly lower in the LG21 strain group than in the placebo group. As salivary amylase concentration increases with sympathetic nervous activity (due to an increase in amylase secretion volume) and decreases with parasympathetic nervous activity (due to an increase in water content of saliva), it can be used as an indicator of autonomic dysfunction under stress. We conducted an exploratory investigation of the relationship between changes in salivary amylase concentration after 12 weeks of intake and baseline salivary amylase concentration. As a result, the effect of continuous intake of the LG21 strain in improving autonomic dysfunction is pronounced in participants with major autonomic dysfunction under stress, and the effect is specific to the LG21 strain (Figure 5). The above results were not enough to support the beneficial effects of LG21 strain on delayed gastric emptying; however, if we define the odds ratio in further study investigated with a larger number of participants, LG21 strain might be expected to have some impact on delayed gastric emptying; further investigation is needed. In addition, during the study, some participants in the placebo group showed improvement in delayed gastric emptying after intake of LG21 strain-free yogurt; further research on yogurt is expected to be conducted.

Even though the LG21 strain was effective for PDS (a subtype of FD) in a previous study [9], there were no statistically significant differences in ΔTmax-AUC during the intake period (the primary endpoint) or the impression score of the overall effect on gastric condition in the PP population. A possible reason for this may be that individuals with mild to moderate delayed gastric emptying were selected as participants, instead of those with gastroparesis, who had severe delayed gastric emptying. As a result, individuals who were not suitable for the aims or assessments of this study (those with normal gastric emptying or those with gastrointestinal motility disorders mainly of the esophagus and small intestine) may have been unexpectedly selected as participants. Therefore, we performed an exploratory gastric symptom-based classification of the participants in this study. First, when we focused on the appetite type of the participants, they were classified into three types: low (43%), medium (36%), and high (21%), as shown in Appendix A. Not only did high-appetite individuals never feel low appetite, but they also had low Tmax, indicating that these participants have few abnormalities related to gastric emptying. Next, we focused on the gastrointestinal symptom types of the participants. For example, it is known that pan-enteric dysmotility, including delayed gastric emptying, is present even in chronic intestinal pseudo-obstruction, mainly comprising small bowel motility disorder [14,21,22]. It has therefore been proposed that delayed gastric emptying should be considered as gastroparesis only when upper gastrointestinal symptoms occur for delayed gastric emptying [1]. As shown in Appendix A, the gastrointestinal symptom types of the participants were classified as PDS-like alone or with vague stomach upset (CLS1, 51.9%), PDS-like with EPS-like or lower abdominal symptoms (CLS2, 22.2%), and PDS-like with gastrointestinal bloating and heartburn (CLS3, 25.9%). The third type was suspected of being pan-enteric dysmotility, and it was therefore decided that it should be treated as a distinct type. Thus, we calculated the odds ratios for improvement in delayed gastric emptying for the efficacy analysis population (PP population) excluding either the high appetite type or the pan-enteric dysmotility type. The odds ratios were 10.7 (95% CI, 1.4 to 82.0) and 9.0 (95% CI, 1.1 to 71.0), respectively, both higher than the odds ratio of 4.1 (95% CI, 0.8 to 20.2) for the PP population. Further, we reanalyzed the participant population excluding both high appetite and suspected pan-enteric dysmotility types, but the odds ratio could not be calculated because no improvement was seen in the placebo group. However, we did find a significant group difference between the placebo group (0%) and the LG21 strain group (80%) (*p* < 0.01, Fisher’s test). The ΔTmax-AUC during the intake period (the primary endpoint) also showed a statistically significant group difference in this population (*p* < 0.05, Figure 6a). Based on the results of the analysis of a strictly defined population of participants with mild to moderate delayed gastric emptying (i.e., participants strongly suspected of having gastric and duodenal disorders), it is thought that continuous intake of the LG21 strain can potentially improve delayed gastric emptying. In addition, in this participant population, impressions of the overall effect on gastric condition showed significant improvement after 6 weeks of intake in the LG21 strain group compared to the placebo group (*p* < 0.05, Figure 6b). Therefore, just as the LG21 strain was effective for PDS in a previous study [9], we consider that continuous intake of the LG21 strain can potentially ameliorate stomach upset in individuals for whom the breath test showed mild to moderate delay in gastric emptying as well. A recent meta-analysis reported a relationship between gastric motility and gastric discomfort [23]. It is also considered that not only delayed gastric emptying but also rapid gastric emptying might be a cause of FD [24]. Rapid emptying into the duodenum is known to be associated with the suppression of gastric mobility and hypersensitivity (the duodenal brake) [25]. Dumping syndrome is a postoperative condition that can develop after esophageal, gastric, or bariatric surgery. It is characterized by abnormalities in the rate of gastric emptying into the duodenum and involves gastrointestinal symptoms (including abdominal pain, diarrhea, borborygmi, nausea, and bloating) caused by indigestion due to rapid gastric emptying [26]. In recent years, it has been suggested that the severity of gastroparesis symptoms is associated with other factors, such as abnormal antropyloroduodenal coordination, rather than delayed gastric emptying [14,27]. The coordinated movement of the stomach and duodenum after a meal is important in terms of digestion and absorption. In particular, the duodenum is known to suppress the gastric emptying rate via hormone secretion and the vagus nerve to control digestion and absorption, for example, to prevent excessive postprandial blood glucose levels [28]. The results of this study indicate the possibility that the continuous LG21 strain intake suppresses the excessive gastric emptying rate (Figure 3c and Appendix A). Therefore, the suppression of the rate of gastric emptying into the duodenum may be involved in the improvement in impressions of the overall effect on the gastric condition of the LG21 strain after continuous intake, but further research is needed.

The mechanism of action of the LG21 strain on gastric motility disorders requires further clarification. However, as mentioned above, the effect of the LG21 strain is prominent in a strictly defined population of participants with mild to moderate delayed gastric emptying (i.e., participants strongly suspected of having gastric and duodenal disorders). Therefore, it is likely that the LG21 strain acts specifically in the upper abdomen, such as the stomach and duodenum. The cause of this may be that the LG21 strain is a lactobacillus that can be expected to have high metabolic activity in the gastric environment due to its intrinsic properties (resistance to artificial gastric juice, ability to grow at low pH, and ability to attach to gastric-derived cultured cells) [29]. The LG21 strain has been detected in the gastric mucus layer [30], and although it does not colonize the stomach, it is believed to temporarily attach; so, with continuous intake, it can be expected to act in the upper digestive tract. In fact, clinical studies have been conducted on the action of the LG21 strain in the upper gastrointestinal tract, and it has been confirmed that the activity of *H. pylori* in the stomach is suppressed by continuous ingestion of the LG21 strain [31,32]. On the other hand, the placebo yogurt also contained the lactobacilli necessary for the fermentation of milk components, and it is possible that they may contribute to improving lower gastrointestinal disorders; however, they may play less of a role in improving upper gastrointestinal disorders. Next, in considering the mechanism of action of the LG21 strain, it is necessary to understand the pathological mechanism of delayed gastric emptying, but it is not yet fully understood. However, reduction in or damage to the interstitial cells of Cajal (ICCs) and nerves by pro-inflammatory macrophages has been revealed in gastroparesis, which is now discussed as a cause of gastric neuromuscular dysfunction [1]. In addition, since the autonomic nervous system plays a significant role in gastric motility, disorders of the autonomic nervous system due to stress and lifestyle disturbances are thought to be among the causes of delayed gastric emptying [7,8]. In recent years, gut-brain interaction has been attracting attention [33], although such interaction involves not only the intestines but also the stomach. Stomach-brain interaction is not only from the brain to the stomach via the efferent vagus nerves but also from the stomach to the brain via the afferent vagus nerves [3,34]. Therefore, there is considered to be a close relationship between abnormalities in the myenteric plexuses of the stomach and abnormalities in the coordination function of the autonomic nerves in the central nervous system. As described above, it is of interest that continuous intake of the LG21 strain caused a significant decrease of salivary amylase concentration, suggesting that continuous intake of the LG21 strain might improve delayed gastric emptying through the improvement of the autonomic nervous function.

However, it is not clear how continuous ingestion of the LG21 strain affects abnormalities in the myenteric plexuses of the stomach and dysfunction in the central/autonomic nervous system. It has been demonstrated that indigenous bacteria in the digestive tract and probiotics stimulate the central nervous system via afferent vagus nerves [35,36,37]; improvement of autonomic function is likely involved in the mechanism of action of the LG21 strain. On the other hand, dysbiosis has attracted attention in the pathogenesis of FD and is thought to increase intestinal mucosal permeability, mainly in the duodenum, and cause inflammation, thereby affecting visceral hypersensitivity and motor function [38,39]. Moreover, there are an increasing number of reports regarding the effects of the LG21 strain and other probiotics on FD [40]. As this study found that the LG21 strain may be effective for mild to moderate delayed gastric emptying, dysbiosis may also be involved in the pathogenesis of delayed gastric emptying. This is interesting because it has been reported that the LG21 strain improves both dysbioses in the upper gastrointestinal tract [41,42,43] and mucosal barrier function [44,45,46]. The stomach has been considered to be less involved in the immune response than the intestine, which is a key component of the immune response. However, a recent study in mice reported that immunity was also induced in the stomach in response to bacteria in the gastric mucosal epithelium [47]. It has also been reported that mucosal immunity is activated by the lactic acid and pyruvic acid produced by intestinal bacteria, albeit in the small intestine [48]. These findings suggest that the LG21 strain may normalize gastric motility by inhibiting inflammation of the submucosal and myenteric plexuses of the stomach, either directly or through the mucosal epithelial flora. However, further research is needed.

This study has several limitations. First, the sample size was small. Second, the sample was skewed toward female participants. This was attributed to the effect of gender differences in mean Tmax in healthy Japanese individuals. In the primary screening, there was a gender difference in the mean Tmax of participants with mild to slightly severe stomach upset (excluding those with suspected FD and diabetics with an HbA1c ≥ 6.5). The mean ± standard deviation was 55.4 ± 10.6 min in women (*n* = 115) versus 47.0 ± 11.4 min in men (*n* = 100), which was a statistically significant difference (Student *t-*test, *p* < 0.001). In this study, the mean Tmax for Japanese individuals was adopted for both men and women to establish criteria for delayed gastric emptying. However, a report has pointed out the need to establish gender-specific criteria that take gender differences into account [49]. Therefore, we investigated gender differences in the effects of the LG21 strain on delayed gastric emptying observed in this study but found no statistically significant difference in the interaction between group and gender. Despite these limitations, these results are based on an RCT. Therefore, the results of this study were not enough to support the beneficial effects of LG21 strain on mild to moderate delayed gastric emptying; however, if we define the odds ratio in further study investigated with a larger number of participants, LG21 strain might be expected to have some impact on mild to moderate delayed gastric emptying. In addition, its effects are likely associated with improved autonomic function in individuals with mild to moderate forms of the disorder, and, to our knowledge, this is the first report of its kind on probiotics. Although it must be ingested continuously, the LG21 strain can be used as a food, which means that it can be ingested daily at the individual’s discretion without relying on medical facilities, making it suitable for daily health maintenance and primary care for those who are concerned about stomach health. In the future, the efficacy and mechanism of action of the LG21 strain need to be investigated with a larger number of participants.

## 5. Conclusions

The results of this study were not enough to support the beneficial effects of LG21 strain on mild to moderate delayed gastric emptying; however, if we define the odds ratio in further study investigated with a larger number of participants, LG21 strain might be expected to have some impact on mild to moderate delayed gastric emptying.

## Figures and Tables

**Figure 1 nutrients-13-01852-f001:**
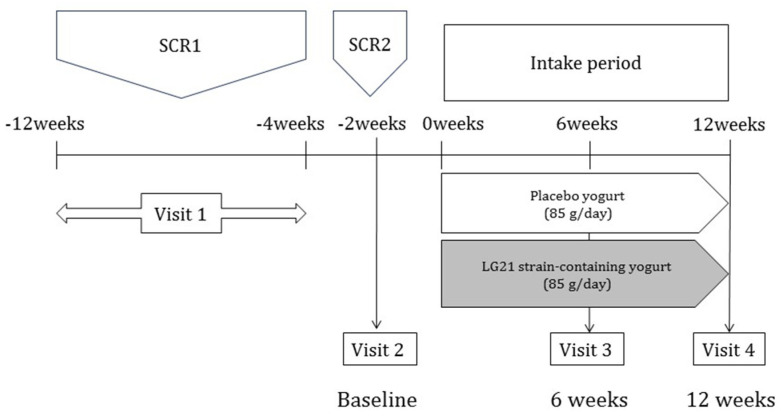
Study design and schedule: The study design and schedule are shown in Figure 1. SCR, screening.

**Figure 2 nutrients-13-01852-f002:**
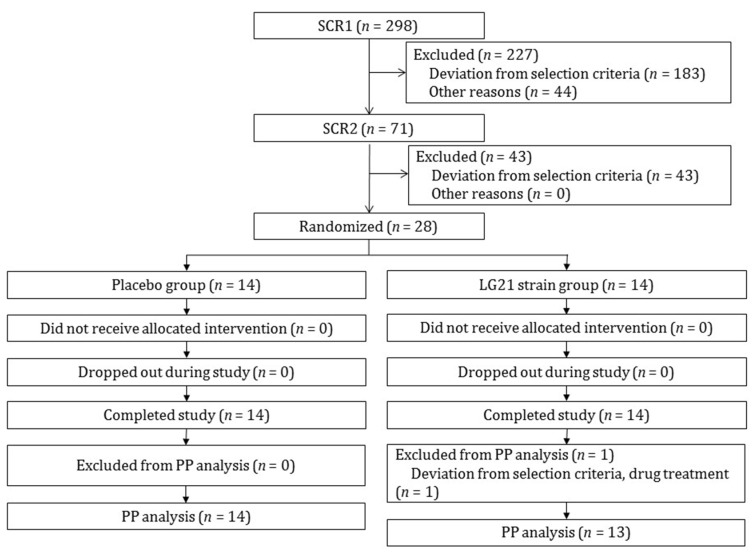
Study flow: The study flow is presented in Figure 2. SCR, screening; PP, per protocol.

**Figure 3 nutrients-13-01852-f003:**
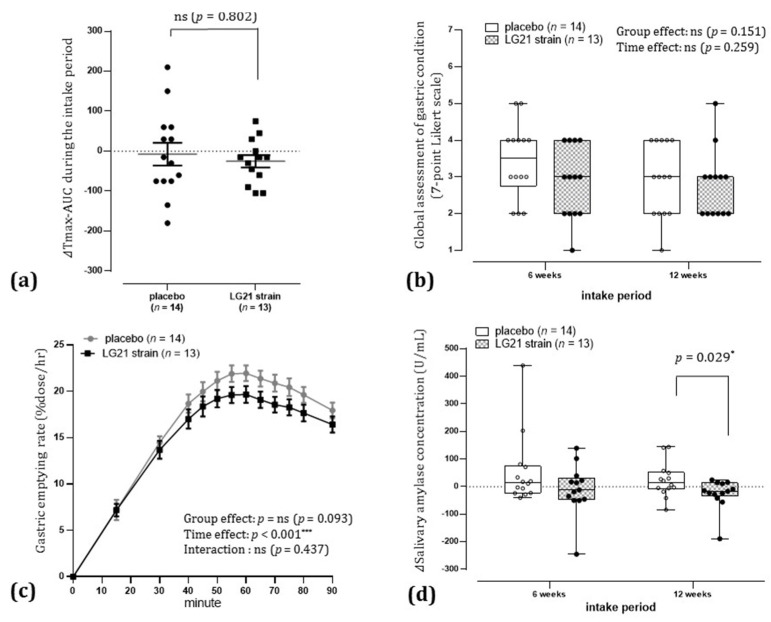
Effects of continuous intake of the LG21 strain in the per-protocol population. (**a**) ΔTmax-AUC during the intake period: The ΔTmax-AUC of each participant was obtained by plotting the origin, ΔTmax after 6 weeks of intake, and ΔTmax after 12 weeks of intake on a graph with intake time (weeks) on the x-axis and ΔTmax (minutes) on the y-axis, and connecting each plot with a straight line to calculate the AUC relative to baseline. The bars indicate the mean and standard error. Intergroup comparison was performed using the Wilcoxon rank-sum test (significance level, *p* < 0.05 [two-sided]). AUC, area under the curve; ns, not significant. (**b**) Global assessment of gastric condition: Participants’ impressions score of the overall effect on gastric condition were shown as dot plots and box plots (median, 25th to 75th percentile; bars show minimum and maximum). Data after 6 and 12 weeks of intake were tested for group effect and time effect using ordinal logistic regression analysis (significance level, *p* < 0.05 [two-sided]). ns, not significant. ns, not significant. (**c**) Gastric emptying rate following ingestion of a liquid meal (after 12 weeks of intake): Gastric emptying rate is shown as mean and standard error. Group effect, time effect and interaction were tested using two-way repeated-measures analysis of variance (significance level, *p* < 0.05 [two-sided]; *** *p* < 0.001 [two-sided]). ns, not significant. (**d**) Δsalivary amylase concentration: Changes in salivary amylase concentrations from baseline are shown as dot plots and box plots (median, 25th to 75th percentile; bars show minimum and maximum). Intergroup comparison was performed using the Wilcoxon rank-sum test (significance level, *p* < 0.05 [two-sided]); * *p* < 0.05 [two-sided]).

**Figure 4 nutrients-13-01852-f004:**
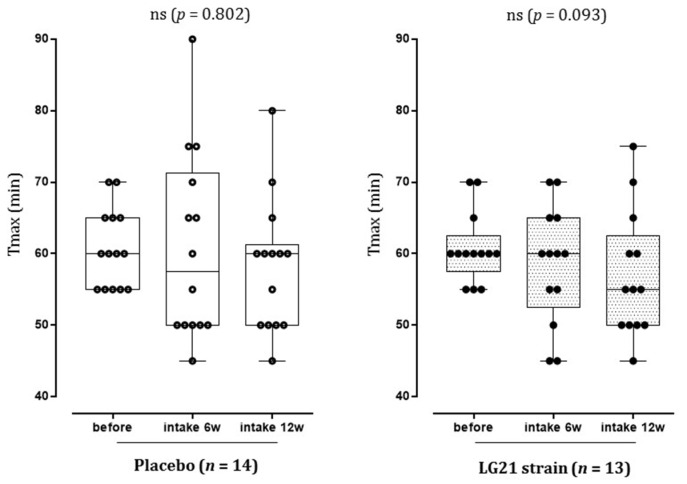
Tmax after 6 and 12 weeks of intake compared to before intake (per-protocol): Tmax after 6 weeks of intake and after 12 weeks of intake compared to before intake for the placebo group and the LG21 strain group are shown as dot plots and box plots (median, 25th to 75th percentile; bars show minimum and maximum). Intragroup comparisons were performed using the Friedman test (significance level, *p* < 0.05 [two-sided]). ns, not significant.

**Figure 5 nutrients-13-01852-f005:**
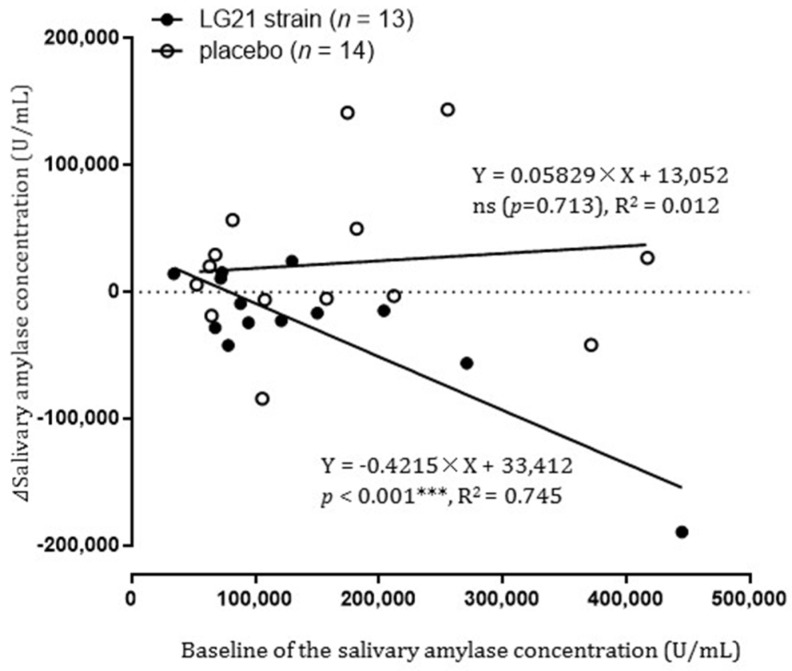
Relationship between salivary amylase concentration at baseline and Δsalivary amylase concentration after 12 weeks of intake (PP population). A scatter plot was created, with salivary amylase concentration before intake on the *x*-axis and Δsalivary amylase concentration after 12 weeks of intake on the *y*-axis, showing regression lines and regression equations for each test food group (significance level, *p* < 0.05 [two-sided]; *** *p* < 0.001 [two-sided]). R^2^, coefficient of determination; ns, not significant.

**Figure 6 nutrients-13-01852-f006:**
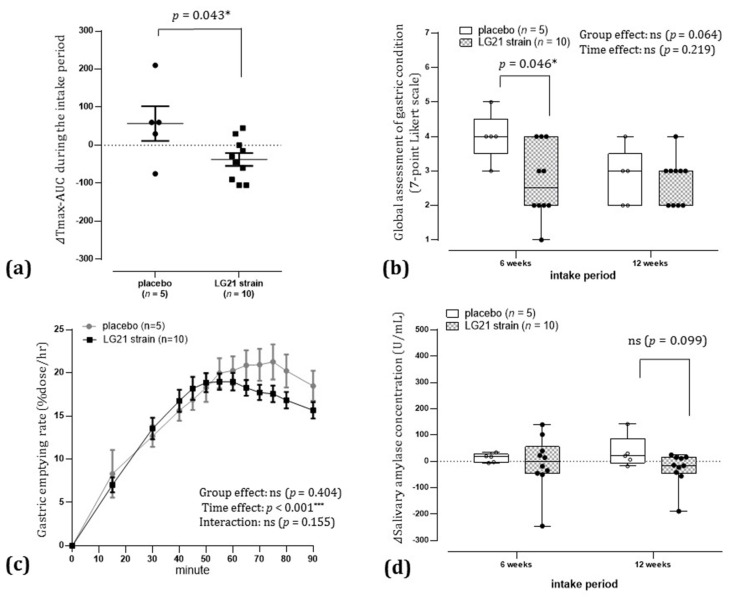
Effects of continuous intake of the LG21 strain in a strictly defined population of participants with mild to moderate delayed gastric emptying (excluding both high appetite and suspected pan-enteric dysmotility types). (**a**) ΔTmax-AUC during the intake period: The bars indicate the mean and standard error. Intergroup comparison was performed using the Wilcoxon rank-sum test (significance level, *p* < 0.05 [two-sided] * *p* < 0.05 [two-sided]). AUC, area under the curve. (**b**) Global assessment of gastric condition**:** Participants’ impressions scores of the overall effect on gastric condition were shown as dot plots and box plots (median, 25th to 75th percentile; bars show minimum and maximum). Data after 6 and 12 weeks of intake were tested for group effect and time effect by ordinal logistic regression analysis (significance level, *p* < 0.05 [two-sided]; * *p* < 0.05 [two-sided]). ns, not significant. (**c**) Gastric emptying rate following ingestion of a liquid meal (after 12 weeks of intake): The gastric emptying rate is shown as mean and standard error. Group effect, time effect, and interaction were tested using two-way repeated-measures analysis of variance (significance level, *p* < 0.05 [two-sided]; *** *p* < 0.001 [two-sided]). ns, not significant. (**d**) Δsalivary amylase concentration**:** Changes in salivary amylase concentrations from baseline are shown as dot plots and box plots (median, 25th to 75th percentile; bars show minimum and maximum). Intergroup comparison was performed using the Wilcoxon rank-sum test (significance level, *p* < 0.05 [two-sided]). ns, not significant.

**Table 1 nutrients-13-01852-t001:** Baseline characteristics of the participants (ITT).

Characteristics		ITT Total (*n* = 28)	Group		*p*-Value
			Placebo (*n* = 14)	LG21 Strain (*n* = 14)	
Age (years), mean ± SD ^†^		41.1 ± 11.0	40.8 ± 9.7	41.4 ± 12.5	0.880
Gender (*n* [%]) ^§^	Female	24 (85.7%)	12 (85.7%)	12 (85.7%)	1.000
	Male	4 (14.3%)	2 (14.3%)	2 (14.3%)	
Classification by BMI (*n* [%]) ^§^	<18.5	4 (14.3%)	2 (14.3%)	2 (14.3%)	1.000
	18.5–25.0	22 (78.6%)	11 (78.6%)	11 (78.6%)	
	≥25.0	2 (7.1%)	1 (7.1%)	1 (7.1%)	
Tmax (min), median (25th–75th percentile) ^‡^		60 (55–65)	60 (55–65)	60 (58.8–65)	0.791
Classification by degree of Tmax (*n* [%]) ^§^	≥55 and <65	19 (67.9%)	9 (64.3%)	10 (71.4%)	1.000
	≥65 and <75	9 (32.1%)	5 (35.7%)	4 (28.6%)	
Severity of stomach upset (*n* [%]) ^§^	Mild	14 (50.0%)	7 (50.0%)	7 (50.0%)	1.000
	Moderate	11 (39.3%)	5 (35.7%)	6 (42.9%)	
	Slightly severe	3 (10.7%)	2 (14.3%)	1 (7.1%)	

Intergroup comparison: ^†^ unpaired Student *t*-test, ^§^ Fisher’s exact test, ^‡^ Wilcoxon rank-sum test. ITT, intention-to-treat; BMI, body mass index.

## Data Availability

The data presented in this study are available on request from the corresponding author.

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
