# Peer review of "The Effect of Continuous Intake of Lactobacillus gasseri OLL2716 on Mild to Moderate Delayed Gastric Emptying: A Randomized Controlled Study"

_nutrients, 2021, doi:10.3390/nu13061852_

Round 1

Reviewer 1 Report

The authors have made relatively minor changes in their revised manuscript. These however, amount to cosmetic changes purely on the discussion and conclusions but, in my opinion, the manuscript is still presented too favourably based on the presented results and the outstanding issues with the lack of statistical power and the authors have not addressed Points 1, 2, 4, 5, and 7 satistifactorily.

Reviewer 2 Report

The paper has been improved according to  request

Author Response

I haven't received any suggestions to answer, so I haven't prepared an answer sheet. Thank you for your supports.

Reviewer 3 Report

Brief abstract: Authors showed for first time the beneficial effect of continuous ingestion of Lactobacillus gasseri OLL2716 strain (LG21) on improving mild and moderate delaying gastric emptying in human volunteers

Highlights: originality, impact in amelioration and/treatment of mild and moderate gastric emptying, robust data analysis, reliable findings supported by congruent parameters.

Discussion:

It would be advisable a more concise and shorter data discussion
It would be advisable include in “Results” data description. For example in lines 431-437, 463-466, 477-481, 486-489, 492-497, 563-568, 663-668, 674-679, describe results therefore should be in “Results” section instead of “Discussion”.

Reviewer 4 Report

This is a well-designed study to test the effects of LG21 in yogurt relative to yogurt placebo on symptoms of delayed gastric emptying. The only potential improvement would have been to design it as a cross-over, but I can see that the interventions were too long to accommodate a cross-over design without risking high levels of subject attrition. So it is the best design possible within the logical constraints of subject compliance and study expenses.

The introduction is interesting and well-written. The methods are carefully described. It is apparent that the statistical analyses were carefully performed and the outcomes of the study were appropriately reported.

Minor Concerns:

Line 20 and Line 27: The number of participants in the study is inconsistent.

Line 131: What were some reasons that subjects were otherwise, “judged by the investigator to be inappropriate for the study”?

Lines 341 – 345: It’s a little confusing to state that gastric emptying rate was “suppressed” with LG21 consumption.  I may be mis-interpreting this, but looking at the graph in Figure 3C, it seems like the % dose/hr (of 13C) detected in the stomach decreased, which indicates that emptying rate INCREASED.

Figure 5: This figure seems unnecessary.  The information available in the text (Lines 334 – 337) is sufficient.

Figures 6 and 7: All analyses that were conducted should be described in the results section.  Therefore, the first description of the data from these two figures (as well as analysis of gender specific effects, Lines 674 - 679) should occur in the results section.

Round 2

Reviewer 1 Report

The authors have continued to make cosmetic changes to the manuscript. At its foundation, there are still methodological issues with the manuscript as detailed in previous reports which have not been satisfactorily addressed by the authors.

Author Response

Thank you for commenting on the review.

The scientific evidence for the conclusions is as detailed in the previous response.

Therefore, there are no plans to revise the manuscript again.

However, we would like to thank you for your many opinions.

This manuscript is a resubmission of an earlier submission. The following is a list of the peer review reports and author responses from that submission.

Round 1

Reviewer 1 Report

Good clinical trial showing a quite new field of appliation of probiotic bacteria

The only weak point is the total absence of a mechanism supporting the outcomes reported in the paper, as it is clear that the strain is unable to reproduce into the gastric enviroment

Reviewer 2 Report

In this manuscript, the authors have completed a randomised, double-blind, and placebo controlled study for the effect of a Lactobacillus gasseri strain on gastric emptying. The study appears to be well designed and conducted. However, based on the reporting of results, this manuscript's conclusions cannot be supported. I have a number of concerns as detailed below, but the overall concern is that the conclusions of the manuscript are too positive towards the use of the L. gasseri strain, particularly in the abstract lines 28 to 34. The authors should thoroughly revised the manuscript so that the conclusions are supported by the presented results (ie: that there is little to no effect of strain supplementation):

  1. Line 143 to 145: this is an inappropriate exclusion criteria as it leads open considerable potential for investigator bias.
  2. No culture confirmation of strain residing in gastrointestinal tract. This is important to show that the LG21 strain is actually capable of colonising the GI tract.
  3. Considerable gender bias in results - which can be a factor in microbiome studies. This should be discussed or only female participants should be used and conclusions limited to this.
  4. If the authors desired 40 participants - was this based on a statistical power test? If so, then by their own calculations the study is under-powered.
  5. For gastric emptying results (Figure 3c) the results do not support a difference between experimental classes. Only the time effect is significant. Did the authors complete an interaction test as in Figure 5?
  6. A P value of > 0.05 is conventionally considered non-significant and the authors should not imply that a P value < 0.10 is significant or trending towards significance. All P values above 0.05 should be labelled ns.
  7. The slicing of the data in Figure 5 is not appropriate and even then, shows no significant group effect nor time x group interaction.